# Electrochemical Detection of *Vibrio cholerae* by Amine Functionalized Biocompatible Gadolinium Oxide Nanoparticles

**DOI:** 10.3390/mi14050995

**Published:** 2023-05-03

**Authors:** Ashutosh Kumar, Tamal Sarkar, Robin Kumar, Amulya K. Panda, Pratima R. Solanki

**Affiliations:** 1Department of Electrical Engineering, University of Notre Dame, Notre Dame, IN 46637, USA; 2Department of Physics, Indian Institute of Technology Kharagpur, Kharagpur 721302, India; 3National Institute of Immunology, New Delhi 110067, India; 4Special Centre for Nanoscience, Jawaharlal Nehru University, New Delhi 110067, India

**Keywords:** Gd_2_O_3_ nanoparticles, *Vibrio cholerae*, immunosensor, biocompatibility

## Abstract

Herein, we report the biocompatible amine-functionalized gadolinium oxide nanoparticles (Gd_2_O_3_ NPs) for the possibility of electrochemical detection of *Vibrio cholerae* (*Vc*) cells. The microwave irradiation process is applied to synthesize Gd_2_O_3_ NPs. The amine (NH_2_) functionalization is carried out via overnight stirring with 3(Aminopropyl)triethoxysilane (APTES) at 55 °C. The size of NPs amine functionalized APETS@Gd_2_O_3_ NPs are determined by transmission electron microscopy (TEM). APETS@Gd_2_O_3_ NPs are further electrophoretically deposited onto indium tin oxide (ITO) coated glass substrate to obtain working electrode surface. The monoclonal antibodies (anti-CT) specific to cholera toxin associated to *Vc* cells are covalently immobilized onto the above electrodes using EDC-NHS chemistry and further BSA is added to obtain the BSA/anti-CT/APETS@Gd_2_O_3_/ITO immunoelectrode. Further, this immunoelectrode shows the response for cells in CFU range from 3.125 × 10^6^ to 30 × 10^6^ and is very selective with sensitivity and LOD 5.07 mA CFUs mL cm^−2^ and 0.9375 × 10^6^ CFU respectively. To establish a future potential for APTES@Gd_2_O_3_ NPs in field of biomedical applications and cytosensing, the effect of APTES@Gd_2_O_3_ NPs on mammalian cells is also observed using in vitro cytotoxicity assay and cell cycle analysis.

## 1. Introduction

In recent years, metal oxide nanoparticles, also known as MONPs, have attracted a great deal of attention due to the possibility that they could be used in biomedical applications [1,2,3]. They are appropriate for use in diagnostics as well as therapy because of their compact size, vast surface area, and ability to target particular cells [4]. Research has been conducted on the potential use of MONPs in imaging, medication delivery, and photothermal therapy [5,6]. They are also capable of performing the role of contrast agents for imaging modalities such as computed tomography scans and magnetic resonance imaging (MRI). In addition to this, it has been demonstrated that MONPs possess antibacterial and anti-inflammatory characteristics [7,8], which positions them as a potentially useful instrument in the battle against infectious diseases and chronic ailments. Because of their one-of-a-kind features, MONPs have emerged as a potentially fruitful area of research for the development of future biomedical applications [9,10,11,12].

In spite of this, the use of MONPs in biological applications is not without its drawbacks, the most notable of which is the potential toxicity of the particles and their instability [13]. Prior to their extensive application in clinical settings, it is essential to perform in-depth analyses of both their safety and their stability [14]. In essence, the utilization of MONPs in biomedical applications possesses a great amount of untapped potential and is a topic of current study and development, with the intention of providing a biocompatible and stable MONP-based platform for enhancing clinical outcomes and overall quality of life for patients [15].

In the field of nanotoxicology, one of the most important areas of research is the investigation of the cytotoxicity of MONPs [16]. Nanoparticles composed of metal oxides have distinctive features that make them appealing for use in a variety of applications in fields such as the biotech industry, medicine, and energy technology. Yet, due to their diminutive size and high surface area-to-volume ratio, they are capable of engaging in interactions with biological systems, including cells, that have the potential to be damaging [17]. Studies of the cytotoxicity of MONPs have shown the impact of these particles on the viability and function of cells. These studies can assist in the identification of potential dangers and provide information regarding the safe utilization of these materials [18]. In order to utilize MONPs, researchers often conduct in vitro tests employing cell culture models to conduct these safety-related investigations. During these experiments, researchers detected changes in cell viability, membrane integrity, and cellular processes [19]. The findings of these investigations have the potential to offer significant insights into the possible effects that MONPs may have on human health and to direct the development of nanomaterials that are less hazardous [20].

Nanoparticles composed of gadolinium oxide (Gd_2_O_3_) have recently attracted a great deal of attention in the field of biomedicine due to the exceptional optical and magnetic properties that they possess [21,22]. In MRI, they can act as contrast agents, and they also have the potential to be used for targeted medication administration, the treatment of hyperthermia, and in vitro imaging [23]. The high relaxivity of Gd_2_O_3_ NPs, which improves the contrast of MRI images and enables better viewing of tissue, is one of the benefits of using these particles [24]. In addition, the fact that they could be less toxic and have a small size makes them excellent for the delivery of drugs to certain cells or tissues in a targeted manner. In order to fully exploit the potential of Gd_2_O_3_ NPs in biomedical applications, additional research will be required [25].

As of now, the toxicity of Gd_2_O_3_ NPs when modified with some other functionalizing agent is still a concern and needs to be carefully evaluated before their widespread use in human subjects. There are some studies that have shown the usage of Gd_2_O_3_ NPs can cause oxidative stress, inflammation, and cell death [23]. Therefore, it is important to optimize synthesis and functionalization methods to minimize their toxicity, so that a method or technique can be developed which can overcome these challenges [26,27]. Despite these challenges, the unique properties of Gd_2_O_3_ NPs have the capability to revolutionize the field of medicine, and further studies are needed to fully understand and harness their potential [28].

Due to the one-of-a-kind optical and magnetic features that Gd_2_O_3_ NPs possess; they play an essential part in the field of biosensing [29]. Additionally, because of the fact that they emit a significant amount of fluorescence, they are frequently utilized as contrast agents in MRI and as sensitizers in other bioimaging applications [24]. In addition to this, they can be functionalized with a wide variety of different biological molecules, which makes it possible to identify particular biomolecules in real time [26,28]. These applications include the detection of cancer, the monitoring of diseases, and the testing of the safety of food.

Some studies suggest that Gd_2_O_3_ NPs can be less toxic and biocompatible, which means that they can be implanted into the body for long-term use without the risk of adverse effects [30]. In addition, the magnetic characteristics of Gd_2_O_3_ make it possible to control and image biosensing phenomena using magnetic fields [29]. Gd_2_O_3_ NPs are valuable for optical imaging and sensing applications due to their high luminescence as well as their high optical stability [22]. In addition, the large surface area of Gd_2_O_3_ NPs offers a vast surface for bio-recognition, which results in increased sensitivity. Because of its characteristics, Gd_2_O_3_ NPs are considered to be a potential material for use in a variety of applications. Some of these applications could be in vitro and in vivo sensing, disease detection, and medication delivery [26].

The present work focuses on the development of 3(Aminopropyl)triethoxysilane coated gadolinium oxide (APTES@Gd_2_O_3_) NP-based immunosensors for the electrochemical detection of bacterial colonies, viz., *Vibrio cholerae* (*Vc*) cells. The APTES@Gd_2_O_3_ -based immunosensor is further applied to detect *Vc* cells in the range of 3.125 × 10^6^ to 50 × 10^6^ CFUs. Prior to this, we have already shown the capability of the developed sensor in the detection of cholera toxin (CT). The developed sensor was not only able to target the CT molecules without any interference but was also reproducible in terms of signal generation [29]. In this work, we challenged our sensor system for more complex detection by employing it for the detection of the bacteria itself directly or indirectly. Further, to establish the bio-compatibility of nanocomposites utilized in the sensor, cytotoxicity and cell cycle studies were performed in this work. The aim was to study the possibility of the usage of these NPs for the development of sensors, which can be used for in vivo sensing. In order to establish the applicability of these materials for implantable device development, the biocompatibility of APTES@Gd_2_O_3_ NPs was evaluated by observing the effect of APTES@Gd_2_O_3_ NPs on mammalian cells, and these are observed using in vitro cytotoxicity assays and cell cycle analysis.

## 2. Experimental Techniques

### 2.1. Materials and Reagents

The following chemicals were obtained from Fisher Scientific: acetonitrile anhydrous 99.8% (CH_3_CN), potassium ferricyanide K_3_ [Fe (CN)_6_], and potassium ferrocyanide K_4_ [Fe (CN)_6_]·3H_2_O. From Sigma-Aldrich, we obtained 3(Aminopropyl)triethoxysilane (APTES), 1-(3-(dimethylamino)-propyl)-3-ethylcarbodimide hydrochloride (EDC) [C_8_H_17_N_3_], and N-hydroxysuccinimide (NHS) [C_4_H_5_NO_3_]. GalaxoLab was the supplier for the sodium chloride (NaCl) salt that was purchased. Merck supplied pellets of sodium hydroxide (NaOH). We obtained sodium monophosphate (NaH_2_PO_4_) and sodium diphosphate dehydrates (Na_2_HPO_4_·2H_2_O) from SRL. CDH was contacted in order to acquire gadolinium (III) nitrate (Gd (NO_3_)_3_·6H_2_O). Antibodies against cholera toxin, Bovine Serum Albumin (BSA), and fetal calf serum were purchased from M/s Genetix Asia Pvt. Ltd. There was no need for any additional purification because all the compounds were of an analytical grade. Na_2_HPO_4_·2H_2_O and NaHPO_4_ were used in the manufacture of phosphate buffer saline (PBS) solutions with a pH of 7.0. De-ionized (DI) water with a resistivity of 18 MΩ cm was used to make a new PBS solution, which was then chilled to 4 °C for storage [29]. DI water was utilized in each and every buffer and solution formulation.

### 2.2. Synthesis of Gadolinium Oxide (Gd_2_O_3_) Nanoparticles

Gd_2_O_3_ NPs were synthesized employing the method used by us in our previous work [29]. Briefly, microwave irradiation was used to create nanoparticles of gadolinium oxide in a single process. During a typical operation, a solution of 1.2 mM gadolinium nitrate was generated by stirring 50 mL of DI water at a temperature of 60 °C for 30 min. Several drops of 5M NaOH were added into the solution dropwise while stirring until the pH of the solution became 10, and again it was agitated for 3 h at 60 °C. After this, the solution was transferred into a container made of borosil glass with a capacity of 100 mL and then placed in a microwave oven of LG 1350 Watt (2450 MHz) for a cycle consisting of 20 s on, 30 s off, and then 10 s on. After completion of the reaction, the vessel was allowed to cool to room temperature. Afterward, the solution was washed with DI water 10 times and then with ethyl alcohol using the centrifuge process 6–7 times (300 rpm) or until the pH of the solution became neutral. The liquid was poured out using a pipette and the resultant slurry was dried at 80 °C for an entire night. For further characterization, this product was put through a mortar and pestle and ground into a fine powder. The product was then annealed at a temperature of 600 °C for three hours. In the end, the result was reduced to a powdery consistency for more in-depth characterization by being ground in a mortar and pestle.

### 2.3. Functionalization of Gd_2_O_3_ with APTES

After dissolving 100 mg of produced Gd_2_O_3_ NPs in 150 mL of propan-2-ol, the mixture was sonicated for fifteen minutes to obtain a highly dispersed suspension. This solution was then agitated at 300 rpm at 55 °C until the nanoparticles dispersed entirely. After that, 1000 mL of APTES was added, drop by drop, to the solution that was prepared earlier, which resulted in the formation of a significant number of amine (-NH_2_) groups on the Gd_2_O_3_ NPs’ surfaces. Finally, 75 mL of DI water was added to the earlier solution. After that, the solution was heated to 55 °C and swirled at a speed of 300 rpm for a total of 72 h. The suspension was centrifuged at 5400 rpm for 20 min in order to remove any unbound APTES by first washing it with DI water and then repeating the operation two or three times. After that, the acquired slurry was dried at a temperature of 50 °C for a period of 48 h, resulting in the production of a whitish material [29].

### 2.4. Electrophoretic Deposition of Functionalized Gd_2_O_3_ on ITO Electrode (APTES@Gd_2_O_3_/ITO)

Electrophoretic deposition (EPD) is a process that uses a DC power source and deposits functionalized nanoparticles of Gd_2_O_3_ on an already hydrolyzed ITO electrode. This technique was used to deposit the nanoparticles. It has been found that EPD is a cost-effective way to make a uniform thin film as compared to other deposition techniques. For this, a colloidal suspension containing 5 mg of APTES@Gd_2_O_3_ was created prior to deposition by ultra-sonification of 0.5 mL of acetonitrile and 0.5 mL of ethanol. This was followed by the preparation of the suspension (30 min). EPD makes use of a voltage source that is either continuous or variable and has two electrode systems. The generation of surface-charged APTES@Gd_2_O_3_ is accomplished through the mixing of magnesium nitrate [Mg (NO_3_)_2_·6H_2_O] salt with the colloidal solution, which functions as an electrolyte. Electrodes were made of platinum foil and ITO glass that had been pre-hydrolyzed. After placing the electrodes one centimeter apart and immersing them in 3 mL of acetonitrile and the colloidal solution that contained APTES@Gd_2_O_3_ nanoparticles, a voltage of 60 volts was applied for a period of ninety seconds. The APTES@Gd_2_O_3_/ITO electrode was measured to have a surface area of 0.25 cm^2^, according to the results. After removing the created electrode from the suspension and washing it with deionized water, the electrode was allowed to dry at room temperature (25 °C) for the next day [29].

### 2.5. Immobilization of Anti-CT onto APTES@Gd_2_O_3_/ITO Electrode

Antibody solution against cholera toxin (anti-CT) at a concentration of 100 µg/mL was freshly prepared in PBS (pH 7.0). The anti-CT solution, EDC (0.2 M), and NHS (0.05 M) were combined in a volumetric ratio of 2:1:1 and held for 30 min before being drop-cast over the APTES@Gd_2_O_3_/ITO electrode. After allowing the electrode to sit in a humidified environment at room temperature for six hours, we cleaned it with PBS (pH 7.0) in order to remove any stray antibody molecules that may have been present. The carboxylic group (-COOH) of anti-CT (Fc’s region) formed a solid amide bond (CO-NH) with the –NH_2_ terminal of APTES@Gd_2_O_3_ NPs. This connection was made between antibodies and nanoparticles with the help of covalent bonding. After that, BSA at a concentration of 0.1 weight percent was used in order to block the nonspecific reactive groups of the electrode. The BSA/anti-CT/APTES@Gd_2_O_3_/ITO immunoelectrode was washed with PBS before it was kept at 4 °C [29]. The fabrication of sensing electrode and biocompatibility has been shown in Figure 1.

### 2.6. In Vitro Cytotoxicity Assay

In addition to this, the cytotoxicity effect of Gd_2_O_3_ NPs on the mammalian cells (RAW264.7) was examined using the MTT assay. In brief, a DMEM-based growth medium containing 10% fetal calf serum was prepared for cell growth and cells were maintained in a humidified environment containing 5% CO_2_ at 37 °C in incubator. For this MTT test, the cells were seeded in 96-well tissue culture plates with 10^4^ cells per well and allowed to grow for 24 h. Following the first 24 h, the medium was changed, and the well-dispersed suspension of Gd_2_O_3_ NPs was added to the cells at concentrations of 25, 50, 100, 200, and 400 μg mL^−1^. The cells were then incubated for another 24 h under the same conditions. The MTT solution was added in each well containing cells after 24 h. At 37 °C, the cells were incubated for a further 3–4 h to allow the MTT dye to be converted into formazan crystals. The development of formazan was visible under an inverted microscope. The solubilization of formazon was carried out in DMSO, and absorbance was measured at 570 nm. The information about cell growth can be calculated from the absorbance values. Relative cell viability was calculated in relation to cells that did not receive Gd_2_O_3_ NP treatment (control 0%) [30].

### 2.7. Cell Cycle Analysis

RAW 264.7 (10^5^) cells per well were seeded in DMEM-containing 12-well plates and incubated for 24 h for cell cycle study. After 24 h, the medium from each well was removed, and cells were treated with 100 and 200 μg mL^−1^ Gd_2_O_3_ NPs. Untreated cells were considered as a control. After 24 h of treatment, the medium was removed, and cells were washed with PBS twice. Then cells were harvested using trypsin, followed by a 5-min period of centrifugation at 1200 rpm. In order to fix the collected cells, chilled 70% ethanol was used. Cells were properly vortexed to prevent clumping and maintained at 4 °C for 1 h. Cells were now centrifuged at a higher speed (2000 rpm) to pellet down. Cells were then washed and resuspended in PBS. The cell suspension was treated with 200 μg mL^−1^ of RNaseA for 10 min, followed by 10 min at 4 °C and dark staining with 50 μg mL^−1^ of propidium iodide. The results of treated cells with Gd_2_O_3_ NPs were compared with the untreated cells.

### 2.8. Instrumentation

The TEM-JEOL 2100F was utilized in order to acquire images of nanoparticles of high resolution. The charge on the surface ofGd_2_O_3_ NPs was determined using a ZEECOM (ZC-2000) Zeta potential analyzer. The electrochemical tests were performed using an AMETEK PARSTAT Potentiostat/Galvanostat. This device has a three-electrode system that consists of a working electrode, an Ag/AgCl electrode that serves as a reference electrode, and a platinum foil counter-electrode. As the electrolyte, a PBS solution with a pH of 7.0 and 0.9% sodium chloride was utilized. The redox agent used was 5 mM [Fe (CN)_6_]^3−/4−^.

## 3. Results and Discussions

### 3.1. Structural and Morphological Study

The physical properties of these NPs were thoroughly studied with the help of X-ray diffraction analysis, UV-Vis spectroscopy, photoluminescence, Fourier transform infrared spectroscopy, scanning electron microscopy, etc., in our previously reported paper [29]. In brief, the Gd_2_O_3_ NPs were characterized using the TEM for structural and morphological characterizations. The TEM sample was prepared using a 2 mg/mL suspension of Gd_2_O_3_ NPs in 70% ethanol solution and ultrasonicated in a bath sonicator for 30 min, which was eventually dropwise-deposited over a copper TEM grid. The structure and shape of Gd_2_O_3_ NPs can be seen in TEM pictures, which show the generation of quasi-spherically structured and mono-dispersed Gd_2_O_3_ NPs, as depicted in Figure 2a. The size of the NPs, on average, was measured to be less than 30 nm. The HRTEM was performed to determine the d spacing in the Gd_2_O_3_ NPs. The d fringes were around 0.313 nm, which corresponds directly to the XRD plane (222) (Figure 2b). The size distribution of nanoparticles ranged anywhere from 18 to 28 nm, as shown in the image (Figure 2c). The charge on the bare Gd_2_O_3_ NPs was estimated by zeta potential. We found that most of the particles were positively charged, and some particles were negatively charged. The charge on the particles may be aroused into them due to friction with each other. The estimated charge on the nanoparticles was 11.26 C.

### 3.2. Electrochemical Characterizations

The electrochemical properties were thoroughly studied in our previous paper of cholera sensor [29]. In brief, the outcomes of cyclic voltammetry (CV) tests that were performed on the following electrodes are presented in Figure 3a: (i) ITO; (ii) APTES@Gd_2_O_3_/ITO; (iii) anti-CT/APTES@Gd_2_O_3_/ITO; and (iv) BSA/anti-CT/APTES@Gd_2_O_3_/ITO. The magnitude of the anodic current, which was measured to be 1.67 mA for the APTES@Gd_2_O_3_/ITO electrode, was found to be significantly lower than that of the bare ITO electrode (2.45 mA) [29]. This occurred as a result of the deposition of APTES@Gd_2_O_3_ materials onto the ITO surface, which slowed the transfer of electrons generated by redox species [Fe (CN)_6_]^−3/−4^ at the electrode/electrolyte interface. This was possible due to the fact that Gd_2_O_3_ NPs were less conductive in their natural state. However, following the immobilization of anti-CT, a rise in current magnitude was seen (2.15 mA). This was due to the fact that there were anti-CT molecules present on the surface of the electrode, which facilitated the transport of electrons between the electrolytes and the electrode surface. These findings suggested that the freely available amine groups of anti-CT made it easier for electrons to be transferred quickly between the electrode and the electrolyte interfaces. In addition to this, the amine group of APTES was responsible for providing a channel by reducing the tunneling distance between the anti-CT electrode and the APTES@Gd_2_O_3_/ITO electrode. The APTES@Gd_2_O_3_/ITO electrode surface served as an appropriate support for the immobilization of biomolecules since APTES molecules were covalently linked to Gd_2_O_3_ NPs and anti-CT. After functionalization of anti-CT/APTES@Gd_2_O_3_/ITO immunoelectrode with BSA, there was a modest increase in the magnitude of current as well as anodic peak potential [29].

Using cyclic voltammetry (CV) at a scan rate of 0.05 V s^−1^ in PBS containing [Fe (CN)_6_]^3−/4−^, we determined the effect of electrolyte pH on the BSA/anti-CT/APTES@Gd_2_O_3_/ITO immunoelectrode (Figure 3b). When the electrolyte pH reached 7.0, the magnitude of the current began to decrease, and it continued to do so after that point. Until that point, the amplitude of the current increased as the pH of the electrolyte increased. At a pH of 7, the peak current was measured to be at its highest value (0.229 mA) [29]. It seems that the BSA/anti-CT/APTES@Gd_2_O_3_/ITO immunoelectrode had a significant amount of activity even when the pH was kept constant. However, the activity of antibodies was rendered ineffective whenever there was a shift in the acidity or basicity of the medium because of the interaction of H^+^ or OH^−^ ions with the amino acid sequence of antibodies. This took place whenever there was a change in the microenvironment.

In order to observe the changes that occurred in the electrochemical properties of the electrodes after antibody immobilization, the study of the electrochemical interface kinetics of the BSA/anti-CT/APTES@Gd_2_O_3_/ITO electrode (Figure 3c) was recorded at different scan rates (10–100 mV s^−1^) during the experiment. Both the cathodic peak current, denoted by *I_pc_*, and the anodic peak current, denoted by *I_pa_*, were shown to be linearly proportional to the scan rate. This linear electrochemical trend demonstrated that the electrochemical reaction is a diffusion-controlled mechanism [29].

The following equations provided the values for the intercepts as well as the slopes.
(1)$$I_{pc\left(BSA-Anti-CT-APTES-Gd_{2}O_{3}-ITO\right)=\left[18.91\;\mu A\left({\mathrm{smV}}^{-1}\right)\times scanrate\left({\mathrm{mVs}}^{-1}\right)+71.42\;\mu A\right],R^{2}=0.998}$$
(2)$$I_{pa\left(BSA-Anti-CT-APTES-Gd_{2}O_{3}-ITO\right)=-\left[18.91\;\mu A\left({\mathrm{smV}}^{-1}\right)\times scanrate\left({\mathrm{mVs}}^{-1}\right)-39.37\;\mu A\right],R^{2}=0.995}$$

Using the equations (Equations (1) and (2)), we can see that the scan rate for the BSA/anti-CT/APTES@Gd_2_O_3_/ITO immunoelectrode is linearly related to the difference between the cathodic and anodic peak potentials (ΔE*_p_* = E*_pc_* − E*_pa_*). Since the data fit well with a linear model, it was assumed that electrons could easily go from the medium to the electrode (Figure 3d) [29].

The cyclic voltametric response of BSA/anti-CT/APTES@Gd_2_O_3_/ITO immunoelectrode towards the detection of CT was already recently reported by our group [29]. In the present study, we challenged our sensor system for more complex detection; here, we employed the sensing of bacteria itself. It has been discovered that CT is also present in the *Vc* cells themselves; therefore, we attempted to detect the bacterium using our sensor system. [31]. The present study focuses on the response of the immunoelectrode towards the electrochemical detection of *Vc* cells directly or indirectly with help of CT. The cyclic voltammograms were recorded for *Vc* cells (3.125 × 10^6^–50 × 10^6^) in PBS carrying [Fe (CN)_6_]^3−/4−^ at a scan rate of 50 mV s^−1^ at room temperature (Figure 4a). For each concentration of *Vc* cells, n = 3 identically prepared immunoelectrodes were taken. A constant decrement in the oxidation peak current value was observed while increasing the concentration of *Vc* cells (Figure 4b).

The decrease in electrochemical response can be due to the immune complex formation of *Vc* cells with the anti-CT present on the immunoelectrode. The immune complex regulates electron transfer at the electrode–electrolyte interface. A linear behavior of oxidation peak current with respect to concentration of *Vc* cells is plotted in Figure 4c. A straight-line fitting gives the linear regression coefficient of the value 0.92. The sensitivity of the BSA/anti-CT/APTES@Gd_2_O_3_/ITO immunoelectrode was evaluated to be 5.07 mA CFUs mL cm^−2^ using the formula slope of the calibration/electrode surface area. The limit of detection (LOD) was calculated to be 0.9375 × 10^6^ CFU using the formula 3 σ/m, where σ represents the standard error of the intercept and m the slope. The above-mentioned values of sensitivity and LOD prove that BSA/anti-CT/APTES@Gd_2_O_3_/ITO immunoelectrode can also serve as an excellent immunosensor for *Vc* cells [29].

The selectivity of the BSA/anti-CT/APTES@Gd_2_O_3_/ITO immunoelectrode was carried out by performing an interference study in the presence of multiple potential analytes in their physiological range, which has been reported earlier. This proved that BSA/anti-CT/APTES@Gd_2_O_3_/ITO immunoelectrode is extremely selective towards *Vc* cells even in the presence of different potential analytes. Therefore, the BSA/anti-CT/APTES@Gd_2_O_3_/ITO immunoelectrode shows significant clinical importance as implantable immunosensors for *Vc* cells detection [29].

### 3.3. In Vitro Cytotoxicity Assay

Using the MTT assay, the cytotoxicity of Gd_2_O_3_ NPs at concentrations ranging from 25 to 400 µg mL^−1^ was assessed on Raw 264.7 (Figure 5). The findings of the MTT experiment showed that the cells are still viable up to 80% even at a greater concentration of Gd_2_O_3_ NPs (200 µg mL^−1^). Our findings suggest that Gd_2_O_3_ NPs within this range may well be utilized for any biomedical application. This suggests that Gd_2_O_3_ NPs can be used for biomedical applications without harming healthy cells, which is a promising conclusion [25,26,28,32,33]. However, more investigation is necessary to examine the long-term effects of exposure to these NPs and to ascertain whether or not these particles are compatible with various types of cells. The results of the MTT assay, taken as a whole, offer useful insights into the potential safety of Gd_2_O_3_ NPs and open the way for further exploration into their use in a number of applications [28].

### 3.4. Cell Cycle Analysis

After treatment with Gd_2_O_3_ NPs, the cell cycle of RAW 264.7 was examined for any alteration in cell cycle phases (Figure 6). Many nano-formulations have been found to have no harmful effects during cytotoxicity studies, although they can alter the cell cycle of treated cells [33]. Cells were categorized into different growth phases for this study. Cells begin in the rest phase G0 and transition into the G1 phase as they gain mass. Cells start the growth phase G2 after replicating their DNA during the S phase. Afterward, cells enter the M phase for cytoplasmic division. Under unfavorable conditions, cells do not follow their normal cell cycle pattern. In this study, cells were treated with 100 and 200 µg mL^−1^ of Gd_2_O_3_ NPs for 24 h. The frequencies of cells in G0-G1, S, and G2-M phases of cell cycle were most similar in the treated cells compared to untreated cells. Our findings showed that cells at these concentrations of NPs were not adversely affected [33]. Together with cytotoxicity tests, it was revealed that Gd_2_O_3_ NPs can be considered biocompatible within the range of (25–200 µg mL^−1^).

## 4. Conclusions

In this work, the possibility of applying BSA/anti-CT/APTES@Gd_2_O_3_/ITO immunoelectrodes as potential sensor for *Vc* cells detection and the applicability of APTES@Gd_2_O_3_ for the development of implantable biosensors, biocompatibility and cell interaction studies were performed. Gd_2_O_3_ NPs were synthesized using microwave irradiation and characterized using several characterization techniques. Further, a thin film of APTES@Gd_2_O_3_ was developed onto ITO using electrophoretic deposition. Antibody against *Vc* cell (anti-CT) was immobilized onto the APTES@Gd_2_O_3_/ITO followed by BSA to fabricate a BSA/anti-CT/APTES@Gd_2_O_3_/ITO immunoelectrode. This immunoelectrode was further used to detect *Vc* cells in the range of 3.125 × 10^6^–50 × 10^6^ CFU. The sensitivity and LOD were evaluated to be 5.07 mA CFUs mL cm^−2^ and 0.9375 × 10^6^ CFU. The biocompatibility of APTES@Gd_2_O_3_ was proved by performing cell viability and cell cycle studies. The effect of APTES@Gd_2_O_3_ NPs on mammalian cells indicated the possibility of its use for biomedical applications. Therefore, the proposed immunosensor can be used as a potential biosensor in future.

## Figures and Tables

**Figure 1 micromachines-14-00995-f001:**
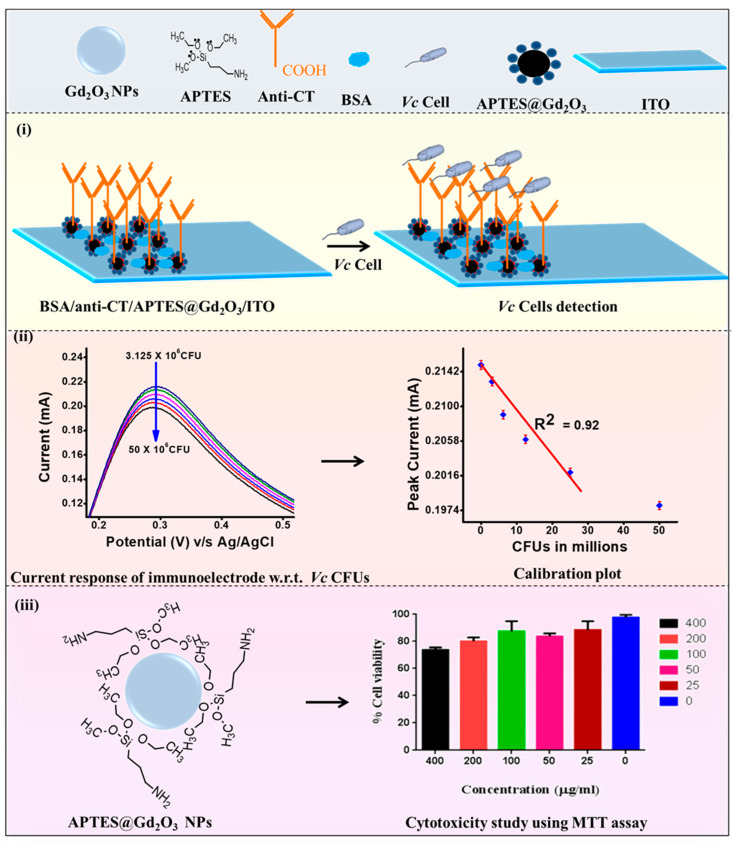
Schematic reparation of the present study; (**i**) immunoelectrode and *Vc* Cells detection, (**ii**) voltametric response of immunoelectrode and calibration plot, and (**iii**) cytotoxicity study of APTES@Gd_2_O_3_ NPs [29].

**Figure 2 micromachines-14-00995-f002:**
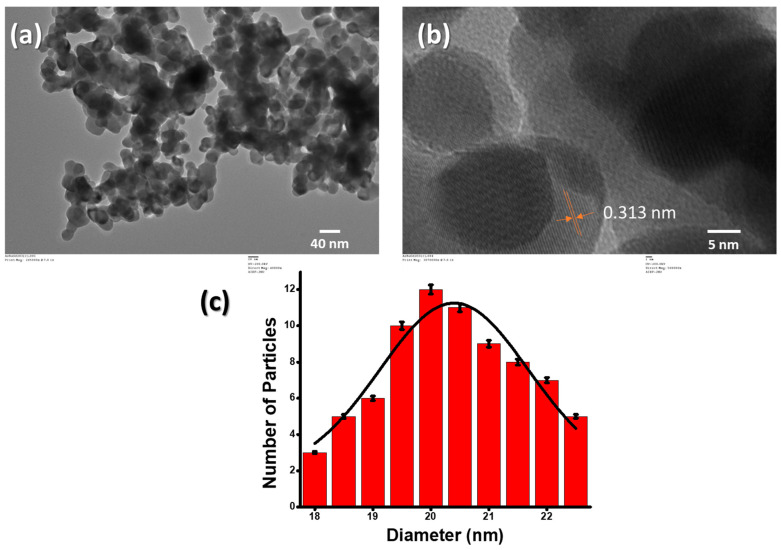
Results of TEM study: (**a**) images of Gd_2_O_3_ NPs; (**b**) HRTEM of Gd_2_O_3_ NPs with crystal planes; (**c**) histogram of particle-size distribution.

**Figure 3 micromachines-14-00995-f003:**
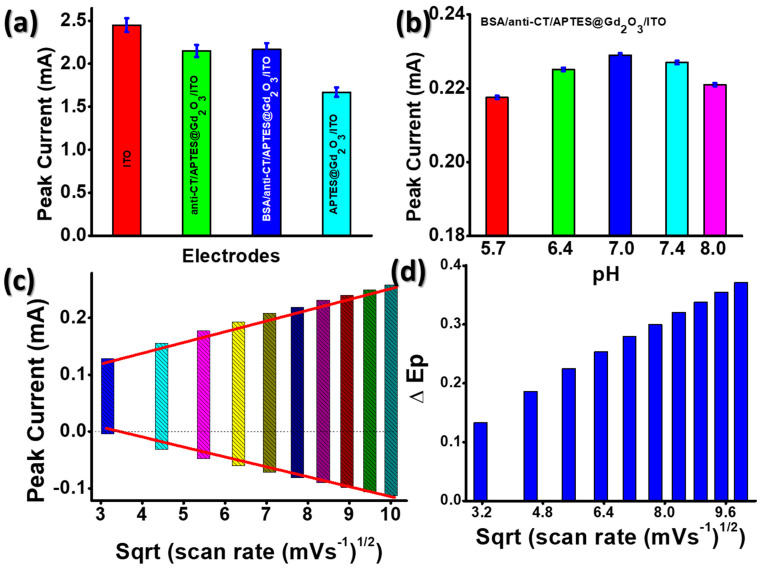
Results of electrochemical study: (**a**) comparison of modified electrodes from oxidation peak current of cyclic voltammogram; (**b**) effect of pH of electrolyte on the immunoelectrode; (**c**) anodic and cathodic peak current variation with increase of sqrt of scan rate varying from 10–100 mV s^−1^; (**d**) variation of difference of anodic and cathodic peak potential with increase of sqrt of scan rate varying from 10–100 mV s^−1^ [29].

**Figure 4 micromachines-14-00995-f004:**
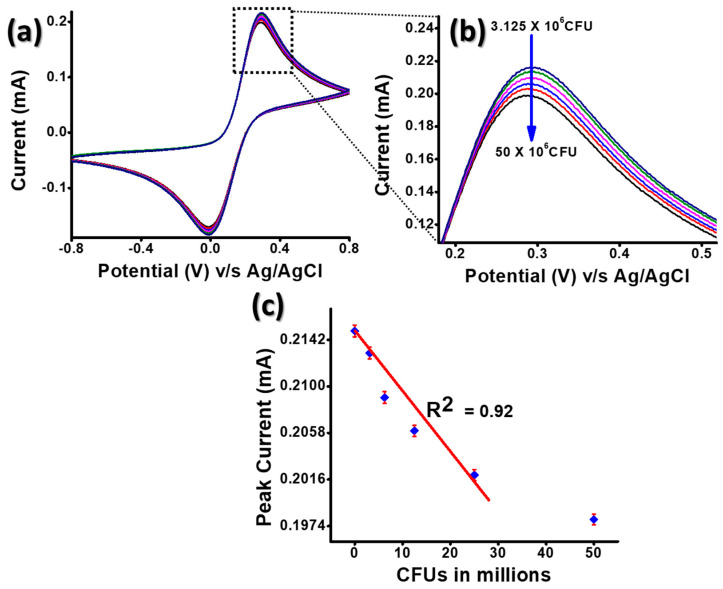
Response study with concentration of *Vc* cells in CFUs: (**a**) cyclic voltametric response of the BSA/anti-CT/APTES@Gd_2_O_3_/ITO immunoelectrode with varying concentration of *Vc* cells ranging from (3.125 × 10^6^–30 × 10^6^); (**b**) enlarged image of the oxidation peak current variation; (**c**) calibration plot of the immunoelectrode plotted from oxidation peak current value for varying concentration of *Vc* cells.

**Figure 5 micromachines-14-00995-f005:**
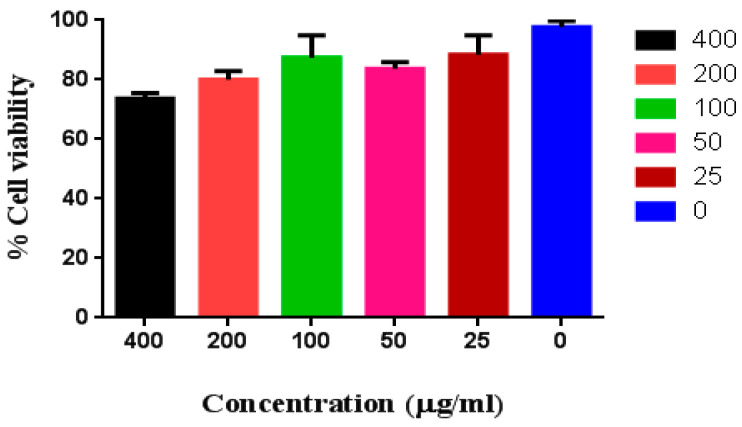
MTT assay in Raw 264.7 cells for Gd_2_O_3_ NPs.

**Figure 6 micromachines-14-00995-f006:**
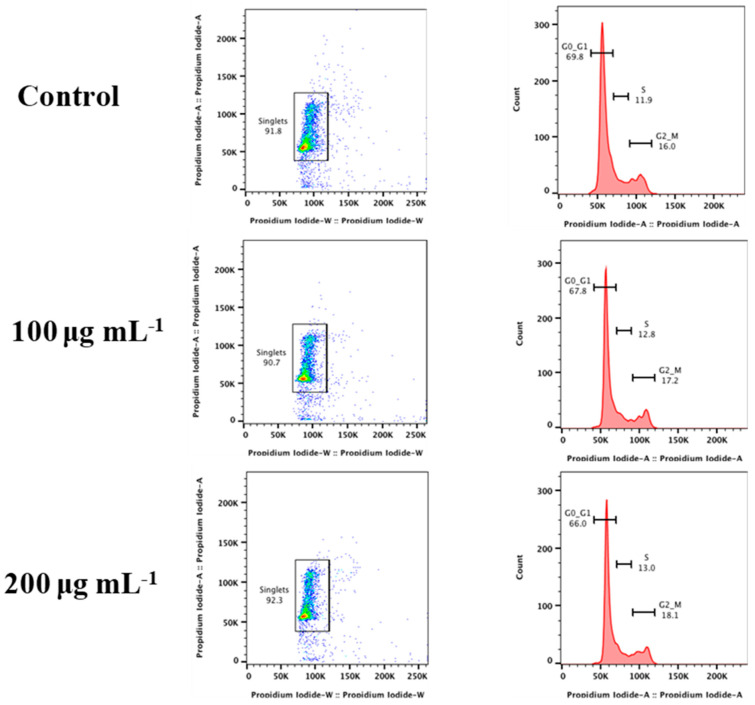
Cell cycle analysis of Gd_2_O_3_ NPs treated RAW 264.7 cells.

## Data Availability

Not applicable.

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
