# Peer review of "Electrochemical Detection of Vibrio cholerae by Amine Functionalized Biocompatible Gadolinium Oxide Nanoparticles"

_micromachines, 2023, doi:10.3390/mi14050995_

Round 1

Reviewer 1 Report

Reviewer Feedback:

Electrochemical detection of Vibrio cholerae by amine functionalized biocompatible gadolinium oxide nanoparticles

In this study, Kumar et al. explored the applicability of Gd2O3 nanoparticles, produced by microwave synthesis and amine functionalized with APTES, for a direct detection of Vibrio cholerae cells. The electrochemical sensor was fabricated utilizing APETS@Gd2O3 deposited on an ITO electrode coated with Cholera toxin-specific monoclonal antibodies. The work, however, serves as an extension of the prior work that the same group of scientists has already published in MDPI's biosensors (Biosensors 2023, 13, 177. https://doi.org/10.3390/bios13020177 ) with a significant degree of overlap. For instance, the procedures for preparing nanoparticles and electrodes are similar. The sole difference is that the prior study detected cholera toxin directly, whereas the current study applies it to Vibrio cholerae cells. In this situation, there are various scientific points that must be thoroughly considered before the paper may be processed further.

-        Authors must include adequate discussion at the end of the introduction to explain the similarities and distinctions between this work and their recently work published in Biosensors, with particular emphasis on the novelty of this work.

-        At the same time, the current form of introduction is lengthy and out of focus. Consider make it more concise. For instance, the paragraphs from lines 65 to 74 and 83 to 91 partially overlap in content. Or, lines 75-77 and 93-95 discussed toxicity in different ways and in different paragraph, which is redundant (one said the toxicity is a concern, while another one said it is non-toxic and biocompatible). To make a manuscript more concise, these areas must be improved, and redundancy must be removed.
I suggest that the authors spend more time on making this introduction part concise and get to the point, especially on the discussion regarding the previous work.

-        I found quite uncomfortable amounts of typos and incorrect use of chemical names and symbols; for instance (but not limited to), K4[Fe(CN)6] is potassium ferrocyanide, not ferricyanide, 3(Aminopropul)triethoxysaline should be (3-Aminopropyl)triethoxysilane instead, the use of DI and D.I., incorrect use of subscript and superscript, e.g., H+, OH- and Ipa. In addition, hydrates is represented by a middle dot, not a dot. Furthermore, analytical grade, not analytical quality, reagents, etc. Additionally, small errors, such as a lack of spacing between words that found randomly in the manuscript convinced me that the authors should put more efforts into proofreading of the manuscript prior further re-submission.

-        Please include additional details to the methodology to enable reproduction of the studies, such as centrifugation condition (RPM, RCF, time, how is the supernatant removed, etc.) Moreover, the requirement for electrophoretic deposition, etc.

-        Raw data of cyclic voltammograms used for Figure 3 should be provided in the SI.

-        The results and discussion section of this manuscript are adequately written and explained, but neither novel nor very compelling, especially with overlapping work with the same group of authors already published; hence, they may not have substantial significance for the discipline.

-        Results in Figure 4c clearly indicate a nonlinear relationship between CFUs and peak current, although the authors attempted to produce a calibration curve using linear regression. This cannot be done, and the authors are required to provide a revised interpretation and discussion for this main result section.

Author Response

In this study, Kumar et al. explored the applicability of Gd2O3 nanoparticles, produced by microwave synthesis and amine functionalized with APTES, for a direct detection of Vibrio cholerae cells. The electrochemical sensor was fabricated utilizing APETS@Gd2O3 deposited on an ITO electrode coated with Cholera toxin-specific monoclonal antibodies. The work, however, serves as an extension of the prior work that the same group of scientists has already published in MDPI's biosensors (Biosensors 2023, 13, 177. https://doi.org/10.3390/bios13020177 ) with a significant degree of overlap. For instance, the procedures for preparing nanoparticles and electrodes are similar. The sole difference is that the prior study detected cholera toxin directly, whereas the current study applies it to Vibrio cholerae cells. In this situation, there are various scientific points that must be thoroughly considered before the paper may be processed further.

-        Authors must include adequate discussion at the end of the introduction to explain the similarities and distinctions between this work and their recently work published in Biosensors, with particular emphasis on the novelty of this work.

Response: Thanks for your valuable suggestion; the manuscript has been edited and distinctions between previous work has been added.

-        At the same time, the current form of introduction is lengthy and out of focus. Consider make it more concise. For instance, the paragraphs from lines 65 to 74 and 83 to 91 partially overlap in content. Or, lines 75-77 and 93-95 discussed toxicity in different ways and in different paragraph, which is redundant (one said the toxicity is a concern, while another one said it is non-toxic and biocompatible). To make a manuscript more concise, these areas must be improved, and redundancy must be removed.
I suggest that the authors spend more time on making this introduction part concise and get to the point, especially on the discussion regarding the previous work.

Response: Thanks for your valuable suggestion; the introduction section of the manuscript has been edited to remove the redundancy.

-        I found quite uncomfortable amounts of typos and incorrect use of chemical names and symbols; for instance (but not limited to), K4[Fe(CN)6] is potassium ferrocyanide, not ferricyanide, 3(Aminopropul)triethoxysaline should be (3-Aminopropyl)triethoxysilane instead, the use of DI and D.I., incorrect use of subscript and superscript, e.g., H+, OH- and Ipa. In addition, hydrates is represented by a middle dot, not a dot. Furthermore, analytical grade, not analytical quality, reagents, etc. Additionally, small errors, such as a lack of spacing between words that found randomly in the manuscript convinced me that the authors should put more efforts into proofreading of the manuscript prior further re-submission.

Response: Thanks for your valuable suggestion; the manuscript has been edited for typo errors; these are caused by different version of document editor software.

-        Please include additional details to the methodology to enable reproduction of the studies, such as centrifugation condition (RPM, RCF, time, how is the supernatant removed, etc.) Moreover, the requirement for electrophoretic deposition, etc.

Response: Thanks for your valuable suggestion; the additional details are added to the manuscript.

-        Raw data of cyclic voltammograms used for Figure 3 should be provided in the SI.

Response: Thanks for your valuable suggestion; we have added CV raw data in the SI section.

-        The results and discussion section of this manuscript are adequately written and explained, but neither novel nor very compelling, especially with overlapping work with the same group of authors already published; hence, they may not have substantial significance for the discipline.

Response: Thanks for your valuable suggestion; the present work is a follow up work of the previously published research, and it is establishing that the materials synthesized using this research can be applied for the development of implantable devices, which may cause any harm to recipient.

-        Results in Figure 4c clearly indicate a nonlinear relationship between CFUs and peak current, although the authors attempted to produce a calibration curve using linear regression. This cannot be done, and the authors are required to provide a revised interpretation and discussion for this main result section.

Response: Thanks for your valuable suggestion; we have edited the manuscript, and clearly indicated the linear relationship in CFUs and current, which is in between 3.125×106 and 30×106 CFU, it was mistakenly mentioned as  3.125×106 to 50×106 CFU.

Reviewer 2 Report

In this interesting work, authors have successfully studied the possibility of applying BSA/anti-CT/APTES@Gd2O3/ITO immunoelectrodes as potential implantable biosensors, biocompatibility, and cell interaction studies.

The acronym APTES, defined in the abstract as 3(Aminopropyl)triethoxysaline (line 14), appears by the first time in the introduction just like APTES, with no definition (line 104). It must be defined again in the introduction. Also, it appears again defined in the Experimental techniques (line 117). This is needlessly.

Figure 2(a) needs a bar indicating the size of the NPs; it appears at the bottom of the picture; however, it is unreadable. All data in the bottom of both pictures (a and b) should be properly included (i.e., readable) in each figure.

Figure 2(c) should include error bars.

Author Response

In this interesting work, authors have successfully studied the possibility of applying BSA/anti-CT/APTES@Gd2O3/ITO immunoelectrodes as potential implantable biosensors, biocompatibility, and cell interaction studies.

Response: Thanks for your valuable comments

The acronym APTES, defined in the abstract as 3(Aminopropyl)triethoxysaline (line 14), appears by the first time in the introduction just like APTES, with no definition (line 104). It must be defined again in the introduction. Also, it appears again defined in the Experimental techniques (line 117). This is needlessly.

Response: Thanks for your valuable suggestion; manuscript has been edited as per your suggestions, however  we want to keep the full name of the chemical in the experimental section so that, it can be easily followed by other researchers for future research.

Figure 2(a) needs a bar indicating the size of the NPs; it appears at the bottom of the picture; however, it is unreadable. All data in the bottom of both pictures (a and b) should be properly included (i.e., readable) in each figure.

Response: Thanks for your valuable suggestion; Figure has been edited as per your suggestions

Figure 2(c) should include error bars.

Response: Thanks for your valuable suggestion; Figure has been edited as per your suggestions

Round 2

Reviewer 1 Report

The current form of the manuscript is adequately edited.